# Differential immunomodulation of T-cells by immunoglobulin replacement therapy in primary and secondary antibody deficiency

Tri Dinh[1,2☯], Jun Oh[1,2☯], Donald William Cameron[2,3], Seung-Hwan Lee[1]*, Juthaporn Cowan [1,2,3]*

1 Department of Biochemistry, Microbiology, and Immunology, Faculty of Medicine, University of Ottawa, Ottawa, Ontario, Canada, 2 Division of Infectious Diseases, Department of Medicine, Faculty of Medicine, University of Ottawa, Ottawa, Ontario, Canada, 3 Clinical Epidemiology Program, Ottawa Hospital Research Institute, Ottawa, Ontario, Canada

☯ These authors contributed equally to this work.
* seunglee@uottawa.ca (SHL); jcowan@toh.ca (JC)

**Data Availability Statement:** All relevant data are within the manuscript.

## Abstract

Patients with primary or secondary antibody deficiency (PAD or SAD) are at increased risk of recurrent infections that can be alleviated by immunoglobulin replacement therapy (IRT). In addition to replenishing antibody levels, IRT has been suggested to modulate immune response in patients with antibody deficiency. Although both commonly treated with IRT, the underlying causes of PAD and SAD vary greatly, suggesting differential modulation of T-cell function that may lead to different responses to IRT. To explore this, peripheral blood mono-nuclear cells (PBMCs) were sampled from 17 PAD and 14 SAD patients before and 2–10 months after initiation of IRT, and analyzed for changes in T-cell phenotype and function. Proportions of CD4, CD8, Treg, or memory T-cells did not significantly change post-IRT compared to pre-IRT. However, we report distinct modulation in T-cell function between PAD and SAD patients post-IRT. Upon α-CD3/CD28 stimulation, proportion of IFN-γ+ CD4 and CD8 T-cells increased in SAD ($p = 0.005$) but not PAD patients post-IRT compared to baseline. Interestingly, total T-cell proliferation was reduced post-IRT in both PAD and SAD patients, although the reduction in proliferation was primarily due to reduced CD4 T-cell pro-liferation in PAD ($p = 0.025$) in contrast to CD8 T-cells in SAD ($p = 0.042$). In summary, even though IRT provides patients with passive humoral immunity-mediated protection in PAD and SAD, our findings suggest that IRT immunomodulation of T-cells is different in T-cell subsets depending on underlying immunodeficiency.

## Introduction

Immunoglobulin (Ig) replacement therapy (IRT) is a blood product therapy prepared from pools of plasma obtained from thousands of healthy blood donors for patients who have inade-quate immunoglobulins, or hypogammaglobulinemia. The treatment can be administered via intravenous (IVIg) or subcutaneous (SCIg) routes, and at a lower dose as compared to the

**Funding:** This study was supported by grants from the Canadian Institute of Health Research http://www.cihr-irsc.gc.ca/e/193.html (MOP-130385 to SHL), and from the University of Ottawa Medicine Research Funds (https://med.uottawa.ca/en/research) to JC. The funders had no role in study design, data collection and analysis, decision to publish, or preparation of the manuscript.

**Competing interests:** Please note that Juthaporn Cowan received research funds and honoraria from CSL Behring, Grifols, Shire and Octapharma, outside the submitted work. This does not alter the authors' adherence to PLOS ONE policies on sharing data and materials.

high-dose Ig use in autoimmunity or inflammatory conditions such as idiopathic thrombocytopenia or chronic inflammatory demyelinating polyneuropathy [1]. For decades, replacement dose IRT has been the mainstay of treatment for patients with inherited (primary) and acquired (secondary) antibody deficiency (PAD and SAD) improving clinical outcomes and preventing recurrent infections [2–7].

Beyond hypogammaglobulinemia in PAD, patients may have dysfunctional and/or deficient T-cell populations, primarily CD4 T-helper cells and regulatory T-cells (Tregs) [8–12]. Moreover, SADs are caused by a heterogeneous group of underlying conditions including but not limited to leukemias/lymphomas, HIV, chemotherapy, malnutrition, corticosteroid use, or other immunosuppressive therapy [13, 14]. Additionally, many SAD patients have conditions that lead to different degrees of impaired or abnormal T-cell function as a result of clinical settings like chronic lymphocytic leukemia (CLL), lymphomas, and B-cell depletion therapy [15–18].

Albeit not completely understood, various immunomodulatory mechanisms of Ig therapy have been previously elucidated *in vitro* and *in vivo* pertaining to both the innate and adaptive immune system [19–21]. Immunomodulatory effect of high dose Ig has been demonstrated as a potential mechanistic efficacy for many inflammatory diseases like Kawasaki disease and myasthenia gravis [19, 22, 23]. Low dose Ig or IRT has also been shown to decrease production of pro-inflammatory cytokines such as IL-2, IL-12, and TNF-α by monocytes in common variable immunodeficiency (CVID) patients [24–26]. However, immunomodulatory effect of IRT toward cell-mediated immunity has not been extensively investigated. A previous study examined the effects of IVIg on cytokine regulation *in vivo* using samples taken before and after replacement-dose (200–400 mg/kg) of IVIg in a group of patients with CVID and X-linked agammaglobulinaemia (XLA) [27]. There was a significant increase in IL-2 expression in CD4 + (and CD4+CD28-) cells and an increase in TNF-α expression in CD8+CD28- cells immediately following IVIg in CVID, but not in XLA patients, while IFN-γ and CD69 expression were not affected by IVIg. In contrast, another study demonstrated that IRT reduced the expression of activated immune markers on T-cells and restored CD4 T-cell counts in CVID [12]. These limited and conflicting data warrant further investigation.

Here, we examined the effect of IRT on T-cell population and function in 31 patients with antibody deficiency, 17 PAD and 14 SAD. It is noteworthy to mention that our 17 PAD cohort is larger than any cohort reported so far for the study of IRT immunomodulation of T-cell function and that the immunomodulatory effects of IRT in patients with SAD has never been studied. Our objectives were to examine the effect of IRT on T-cell population and function in both PAD and SAD patients. By examining proportions, cytokine production, and proliferative potential of PBMCs from the patients, we identified that IRT induces differential immunomodulatory effects on T-cells between patients with PAD and SAD.

## Materials and methods

### Study patients

Patients with hypogammaglobulinemia were recruited from the Immunodeficiency Clinic at the Ottawa Hospital General Campus between 2013 and 2018, and stratified into primary (1°) or secondary (2°) antibody deficiency. Inclusion criteria are decreased IgG level, eligible for and agree to receive IRT, ability to provide informed consent, and availability for ongoing follow-up. Definitive diagnosis of hypogammaglobulinemia is characterized as a serum IgG of below 7g/L. Patient demographic data regarding age, sex, weight, underlying immunodeficiency, comorbidities, and current medication was noted during the study. Data on IRT dosage (g/kg), route of administration (SCIg or IVIg), and duration of treatment was also noted.

IVIg was administered in the hospital every 3–4 weeks while SCIg was self-administered at home once to twice per week. Baseline IgG, IgA, and IgM were measured by nephelometry methods while IgG subclasses (IgG1, IgG2, IgG3, IgG4) were measured by electrophoresis in a clinical laboratory. History of recurrent infections was not explicitly stated in our study protocol. However, patients who were referred to the clinic for consideration of IRT generally have histories of recurrent or severe infections. This study protocol was approved by the Ottawa Health Science Network Research Ethics Board (protocol ID: 20130310-01H).

## Isolation of peripheral blood mononuclear cells (PBMCs)

Approximately 40mL of blood was drawn before and after at least 8 weeks post-IRT in heparinized tubes (BD Biosciences, NJ, USA) Blood was processed into peripheral blood mononuclear cells (PBMCs) via Ficoll gradient separation (GE Healthcare, MA, USA) and subsequently cryopreserved at -80˚C in 90% FBS + 10% DMSO. PBMCs were later thawed in batches and rested overnight cultured in RP-10 media (RPMI 1640 (Gibco) supplemented with 10% FBS, 50 µM 2-mercaptoethanol (Gibco), 100µg/mL penicillin (Gibco), 100µg/mL streptomycin (Gibco), 10mM HEPES (Gibco), and 10 mM L-glutamine (Wisent, QC, Canada)) before use in assays at 37˚C and 5% $CO_2$.

## Flow cytometry and antibodies

Antibodies used for flow cytometric analysis are as follows: α-CD3-BV510, α-CD4-PE-Cy7, α-CD8-BV786, α-CD45RO-BV650, α-IFNγ-FITC, α-TNFα-APC from BD Biosciences, α-CD25-PE from BioLegend (CA, USA), α-CD127-AF700, α-FOXP3-APC from ThermoFisher (MA, USA), and Live/Dead-APC-Cy7 from Life Technologies (CA, USA). Samples were acquired via flow cytometry with a LSR Fortessa Cell Analyzer (BD Biosciences). Subsequent gating of flow cytometry data was performed with Kaluza 1.3 software (Beckman Coulter, CA, USA).

## Assessment of T-cell phenotype

Thawed PBMC were stained for surface expression of CD3, CD4, CD8, CD25, CD45RO, and CD127, followed by fixation in 2% PFA and stained for intracellular expression of FOXP3 using the FOXP3 transcription factor buffer staining kit as outlined by the manufacturer (ThermoFisher). Phenotyping for CD4+ T-cells, CD8+ T-cells, Tregs, naïve (CD3+CD45RO-), and memory (CD3+CD45RO+) T-cells was then performed via flow cytometry. Tregs were defined as the CD3+CD4+CD25hi+CD127lo+FOXP3+ gated population [28, 29].

## T-cell activation assays

For T-cell stimulation, 96-well U-bottom culture plates (Corning, NY, USA) were coated overnight at 4˚C with 1µg/mL α-CD3 antibodies (ThermoFisher) and washed twice with 1x PBS. $2 \times 10^6$ thawed PBMCs in RP-10 media were then stimulated with 2µg/mL soluble α-CD28 antibodies (BD Biosciences) for 6 hours. For T-cell activation by CEF (cytomegalovirus, Epstein-Barr virus, and influenza virus) combo peptides, thawed PBMCs in RP-10 media were activated in 96-well U-bottom culture plates (Corning) with 1µg/mL of CEF combo peptides (NIH AIDS Reagent Program, MD, USA) for 6 hours. α-CD3/CD28 stimulation is MHC-independent, allowing for a stimulation signal for bulk T-cells in PBMCs while CEF peptides are a stimulation for a viral specific memory CD8 T-cells [30, 31].

## Assessment of cytokine production

To measure extracellular cytokine production from whole PBMCs, thawed PBMCs were activated with α-CD3/CD28 for 6 hours for IFN-γ assay and 48 hours for IL-10 and IL-17-α assays. Supernatant was obtained for cytokine detection by flow cytometry using the BD Cytometric Bead Array Kit (BD Bioscience) as outlined by manufacturer protocol, with data analysis performed using FCAP Array Software v3.0 (BD Biosciences). To assess intracellular cytokine production of T-cells in PBMCs, 5μg/mL brefeldin A (BFA) (Sigma-Aldrich, MI, USA) was added to the anti-CD3/CD28 or CEF peptide stimulated PBMCs for the last 5 hours of stimulation. Fixation and permeabilization was performed using BD Cytofix/Cytoperm kit according to the manufacturer protocol (BD Biosciences) and intracellular cytokines were stained with fluorochrome conjugated α-IFN-γ and α-TNF-α antibodies for 25 minutes after permeabilization.

## Assessment of T-cell proliferation

To assess T-cell proliferation, thawed PBMCs were stained with CellTrace dye as per manufacturer protocol (ThermoFisher) and activated with α-CD3/CD28 for 4 days. Cells were subsequently surface stained for CD3, CD4, and CD8, followed by fixation in 2% paraformaldehyde (PFA) before flow cytometry acquisition. Cell proliferation index was defined as the inverse of the median fluorescence intensity (MFI) of the gated proliferated population times 100. This is a slightly modified adaptation of common MFI interpretation where a lower MFI represents a higher degree of cell dye dilution and thus a higher cell proliferation index [32, 33].

## Statistical analysis

Statistical analyses were performed on Graphpad Prism 8.0.0 (GraphPad Software Inc, CA, USA). Statistical comparisons of patients pre- and post-IRT was performed using the Wilcoxon matched-pairs signed rank test. Differences were considered to be statistically significant when $p < 0.05$, indicated by $^*$ $p < 0.05$ and $^{**}$ $p < 0.01$.

## Results

### Patient characteristics

We obtained paired peripheral blood samples from 31 patients with antibody deficiency, 17 were primary and 14 were secondary (Table 1). The PAD patients were younger than the SAD with mean age of 49.3±14.7 vs 63.9±15.1. There were more female in the PAD group (82.4%) while male was slightly more in the SAD group (57.1%). The most common disease in PAD was CVID (64.7%), followed by subclass deficiency (35.3%). Causes of SAD cases were mostly hematological malignancies including chronic lymphocytic leukemia (CLL) (42.9%), non-Hodgkin Lymphoma (NHL) (28.6%), and multiple myeloma (MM) (7.1%). Chronic lung diseases were common comorbidities in both groups. None of the PAD cases were taking systemic immunosuppressive therapy. In contrast, many SAD patients were receiving systemic corticosteroids (n = 3), rituximab (n = 3), ibrutinib (n = 1), and non-steroid immunosuppression (n = 3). Eight patients (57.1%) also had history of past rituximab treatment. The majority of patients received SCIg (94.1% in PAD, and 100% in SAD). Mean dosage was 0.15±0.05 and 0.12±0.04 g/kg/week in PAD and SAD, respectively. Baseline immunoglobulin levels were similar in both groups (Table 1). IRT normalized IgG levels but not IgA or IgM as expected. Absolute cell count of lymphocyte subsets were largely normal in PAD pre-IRT (Table 1). Clinical absolute blood cell count data for SAD was incomplete as T/B/NK enumeration testing was

**Table 1. Characteristics of antibody deficiency patients.**

| | Primary (1°) Antibody Deficiency (PAD) (n = 17) | Secondary (2°) Antibody Deficiency (SAD) (n = 14)* |
|---|---|---|
| **Mean age (SD)** | 49.3 (14.7) | 63.9 (15.1) |
| **Sex (M:F)** | 3:14 | 8:6 |
| **Associated disease** | • CVID (64.7%)<br>• Subclass deficiency (35.3%) | • CLL (42.9%)<br>• NHL (28.6%)<br>• MM (7.1%)<br>• Solid tumor (7.1%)<br>• Kidney transplant (7.1%) |
| **Comorbidities** | | |
| • Diabetes | 1 | 1 |
| • Chronic lung diseases | 7 | 3 |
| • Cirrhosis | 1 | 0 |
| • Chronic kidney diseases | 0 | 1 |
| • Autoimmune diseases | 5 | 0 |
| **Medications** | | |
| • Systemic corticosteroids | 0 | 3 |
| • Inhaled steroids | 5 | 5 |
| • Non-steroid immunosuppressant | 0 | 3 |
| • Current rituximab treatment | 0 | 3 |
| • Previous rituximab treatment | 0 | 8 |
| • Other immunobiological treatment | 0 | 1 |
| **Mean immunoglobulin levels pre-IRT** (g/L) | | |
| • IgG (SD) | 3.49 (2.19) | 3.89 (1.59) |
| • IgA (SD) | 0.47 (0.64) | 0.61 (0.76) |
| • IgM (SD) | 0.46 (0.48) | 0.31 (0.30) |
| **Mean immunoglobulin levels post-IRT** (g/L) | | |
| • IgG (SD) | 9.98 (2.59) | 9.79 (1.92) |
| • IgA (SD) | 0.48 (0.67) | 0.55 (0.52) |
| • IgM (SD) | 0.61 (0.72) | 0.27 (0.26) |
| **Baseline lymphocyte subset** (cells/μL) | | |
| • CD3 (SD) | 1326 (416) | 1035 (668)* |
| • CD4 (SD) | 911 (299) | 528 (299)* |
| • CD8 (SD) | 367 (178) | 486 (367)* |
| • CD4/CD8 ratio (SD) | 3.01 (1.53) | 1.41 (0.67)* |
| • CD19 (SD) | 239 (235) | 113[§] / 13041[‡] (31702)*[‡] |
| • CD16+CD56+ (SD) | 186 (95) | 175 (137)* |
| **CMV serostatus** | 1/16 (6.3%) | 2/7 (28.6%) |
| **EBV serostatus** | 14/15 (93.3%) | 5/6 (83.3%) |
| **IRT** | | |
| • IVIg: SCIg | 1:16 | 0:14 |
| • Mean Dosage g/kg/week (SD) | 0.15 (0.05) | 0.12 (0.04) |
| **Mean duration between pre and post-IRT blood samples in weeks (SD)** | 21.6 (10.8) | 27 (18) |

*. Baseline lymphocyte subset analysis of patient blood was only performed for a minority of SAD patients (6/14), as it was not deemed within the necessary clinical standards of diagnostic care for SAD alone.

‡. Value is skewed due to one patient with severe polycythemia + CLL.

§. Median is shown in addition to mean.

not routinely requested for SAD in our clinical practice, leading to incompletely represented lymphocyte subset data.

## IRT does not appear to alter T-cell phenotype proportions

CD4 T-cell populations have been described to be reduced in CVID patients, while CD8 T-cell populations remain unchanged, resulting in a high rate of inverted CD4/CD8 ratios (<1.0) among CVID patients [34, 35]. Ranges for normal T-cell subset proportions in PBMC have been reported in a large epidemiological study of healthy individuals for CD4 (29%-63%, 49% mean), CD8 (19%-38%, 28% mean), and CD4/CD8 ratio (0.83–3.04, 1.83 mean) [36]. To determine whether IRT modulates proportions of T-cell compartments, we examined proportions of CD4, CD8, and memory T-cells in PBMC collected pre-IRT and post-IRT (Fig 1A). Pre-IRT, we observed a high degree of heterogeneity in T-cell proportion in PBMC for CD4 (6%-64%, 42% mean), CD8 (2%-35%, 15% mean), and CD4/CD8 ratio (0.75–7.3, 3.4 mean) in both PAD and SAD patients (Table 2). While some patients did reflect healthy T-cell proportions, many were below healthy ranges for CD4 and CD8. Post-IRT, we found that CD4 and CD8 T-cell proportion and CD4/CD8 ratio did not significantly change, and further subanalysis into memory CD45RO$^+$ T-cell proportion also did not reveal any significant changes post-IRT (Fig 1B–1F). Substratification into PAD and SAD patients also did not yield a change in either group. It is important to note that unlike the incomplete whole blood enumeration for SAD, PBMC data was available for all PAD and SAD patients and thus is representative of the entire patient cohort.

In interest of the key suppressive role in adaptive immune system regulation of effector T-cells, we analyzed changes in Treg proportion post-IRT, as previously defined by other groups as the CD3$^+$CD4$^+$CD25$^{hi}$CD127$^{lo}$FOXP3$^+$ population [29]. Pre-IRT, we report a broad range of Tregs as a proportion of CD4 T-cells (0.8%-9.4%, 1.9% mean) (Table 2). Post-IRT, we report no significant changes in the Treg proportion in all hypogammaglobulinemia patients (Fig 1E). Changes were not observed in the subgroup of PAD or SAD patients either.

Interestingly, 2 of the 31 patients exhibited elevated CD4/CD8 ratios up to 7.5 pre-IRT. There was no change in CD4/CD8 ratio post-IRT initiation and only one patient had inverted CD4/CD8 ratio. Baseline proportion of CD8 T-cell in total PBMCs in our patient cohort appeared to be in the lower range as compared to those of healthy individuals from a large epidemiological study while CD4 T-cell proportion appeared to be within the normal range [36]. Therefore, the elevated CD4/CD8 ratios were likely due to decreased CD8 T-cell rather than an increased CD4 T-cell proportion. However, our data cannot exclude the possibility of a mild degree of decreased absolute CD4 and large degree of decreased absolute CD8 cell counts to explain an elevated CD4/CD8 ratio because we measured T-cell proportion in PBMCs instead of whole blood. Nevertheless, this possibility was less likely as the baseline lymphocyte counts obtained from the clinical laboratory reported relatively normal CD4 and CD8 T-cell counts (Table 1). In summary, we report abnormal perturbations in CD4/CD8 ratios of PBMCs in PAD and SAD patients with no significant changes post-IRT.

## Effect of IRT on T-cell function and cytokine production between PAD and SAD

Helper T (Th) cells are divided into subsets which modulate the adaptive immune system to induce distinct immune responses. To investigate whether IRT induces altered function of helper T (Th) cell subsets, we analyzed production of IFN-γ, IL-10, and IL-17α, representative cytokines of Th1, Th2, and Th17 respectively [28, 37, 38]. Using a cytometric bead array (CBA), we measured the cytokine concentration of IFN-γ, IL-10, and IL-17α in the

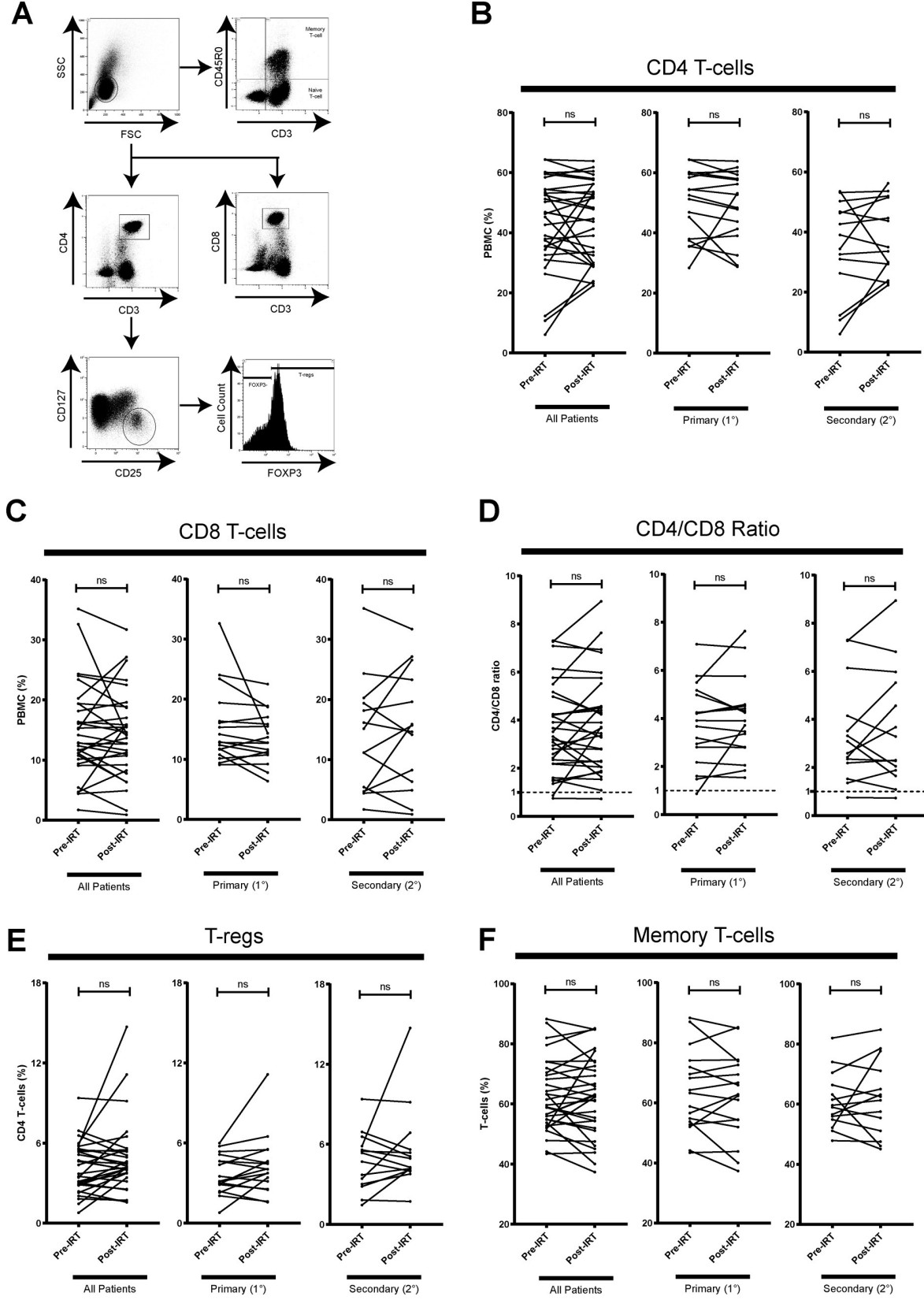

**Fig 1. Comparison of T-cell subset population proportions of unstimulated PBMCs pre- and post-IRT.** (A) Thawed cryopreserved patient PBMCs were stained for CD3, CD4, CD8, CD25, CD45RO, CD127, and FOXP3 for T-cell phenotyping. Gating strategy defines CD4$^+$ T-cells, CD8$^+$ T-cells, Tregs (CD3$^+$CD4$^+$CD25$^{hi}$CD4$^{lo}$FOXP3$^+$), and memory T-cells (CD3$^+$CD45RO$^+$). Proportions of CD4$^+$ T-cells (B), CD8$^+$ T-cells (C), CD4/CD8 ratio (D), Tregs (E), or memory T-cells (F) in PBMC do not significantly change post-IRT. Patients are stratified into 1˚ (n = 17) and 2˚ (n = 14) antibody deficiency. ns denotes not significant ($p > 0.05$) and $p$-values were determined by Wilcoxon matched-pairs signed rank test.

supernatant of α-CD3/CD28 stimulated PBMCs (Fig 2). Analyzing the patients as a whole or stratifying the patients into PAD and SAD patients did not reveal significant change in the concentration of IL-10, or IL-17α post-IRT (Fig 2B and 2C). IFN-γ cytokine concentration in supernatant was not found to significantly change post-IRT in PAD patients nor SAD patients (Fig 2A). Since IFN-γ is known to be secreted mainly by CD4 T-cells, CD8 T-cells, and NK cells, we cannot discern the specific immune cell populations that are expressing these cytokines in whole PBMCs though supernatant cytokine measurement alone. To address this, we used intracellular cytokine staining of IFN-γ and flow cytometry to investigate changes in cytokine expression in CD4 and CD8 T-cell subsets after 6 hr of α-CD3/CD28 stimulation (Fig 3A). There was a moderate proportion of IFN-γ+ CD4 T-cell (0.6% - 4.6%, 2.2% mean) and a strong proportion of IFN-γ+ CD8 T-cell (1.3% - 39%, 9.4% mean) after α-CD3/CD28 stimulation. We found that PAD patients did not exhibit a significant change in proportion of IFN-γ + CD4 or IFN-γ+ CD8 T-cells post-IRT (Fig 3B and 3C). On the other hand, SAD patients showed higher proportion of IFN-γ+ CD4 ($p = 0.04$) and IFN-γ+ CD8 ($p = 0.005$) T-cells. CMV, EBV, and influenza are common viruses that most adults have encountered. In our cohort, most patients had evidence of previous EBV infection (Table 1). To examine the effect of IRT on viral specific memory CD8 T-cells, CEF (cytomegalovirus, Epstein-Barr virus, and influenza virus) peptide activated PBMCs were assessed for intracellular IFN-γ and TNF-α expression in T-cells pre- and post-IRT (Fig 4A), as TNF-α is another key effector cytokine for CD4 and CD8 T-cells [39, 40]. The proportion of IFN-γ and TNF-α double-positive CD8 T-cells was low (0.04% - 5.9%, 1.2% mean) and there was no significant change in expression found in patients post-IRT (Fig 4B), with further substratification of patients into PAD and SAD revealing no significant differences. Single-positive expression of IFN-γ or TNF-α in CD4 or CD8 T-cells was also not significantly different post-IRT (Fig 4C and 4D). Taken together, our data indicated that potential to produce IFN-γ of CD4 and CD8 T-cells upon the TCR stimulation is enhanced post-IRT. Nonetheless, improved functionality of the memory T-cell compartment was not observed based on the expression of IFN-γ or TNF-α upon stimulation with common viral antigens.

**Table 2. T-cell subsets of antibody deficiency patients pre-IRT.**

| T-cell Subset | Proportion of PBMC | | | | | | | | |
| | All Patients (n = 31) | | | Primary (1˚) Antibody Deficiency (n = 17) | | | Secondary (2˚) Antibody Deficiency (n = 14) | | |
| | Mean | SD | Range | Mean | SD | Range | Mean | SD | Range |
|---|---|---|---|---|---|---|---|---|---|
| CD4 | 42.5 | 15.6 | 6.1–64.4 | 46.1 | 15.8 | 12.3–64.4 | 38.4 | 14.8 | 6.1–53.6 |
| CD8 | 14.8 | 8.1 | 1.7–35.2 | 15.1 | 8.3 | 1.7–32.6 | 14.4 | 8.3 | 4.4–35.2 |
| CD4/CD8 ratio | 3.56 | 1.82 | 0.75–7.31 | 4.00 | 2.11 | 0.87–7.31 | 3.05 | 1.32 | 0.75–6.14 |
| Treg* | 4.2 | 1.9 | 0.8–9.4 | 3.7 | 1.5 | 0.8–6 | 4.8 | 2.2 | 1.5–9.4 |
| Memory T-cell‡ | 61.2 | 14.7 | 26.1–88.3 | 60.0 | 18.6 | 26.1–88.3 | 62.5 | 9.0 | 47.8–79.6 |

*: Proportion of CD4 T-cells.

‡. Proportion of CD3 T-cells.

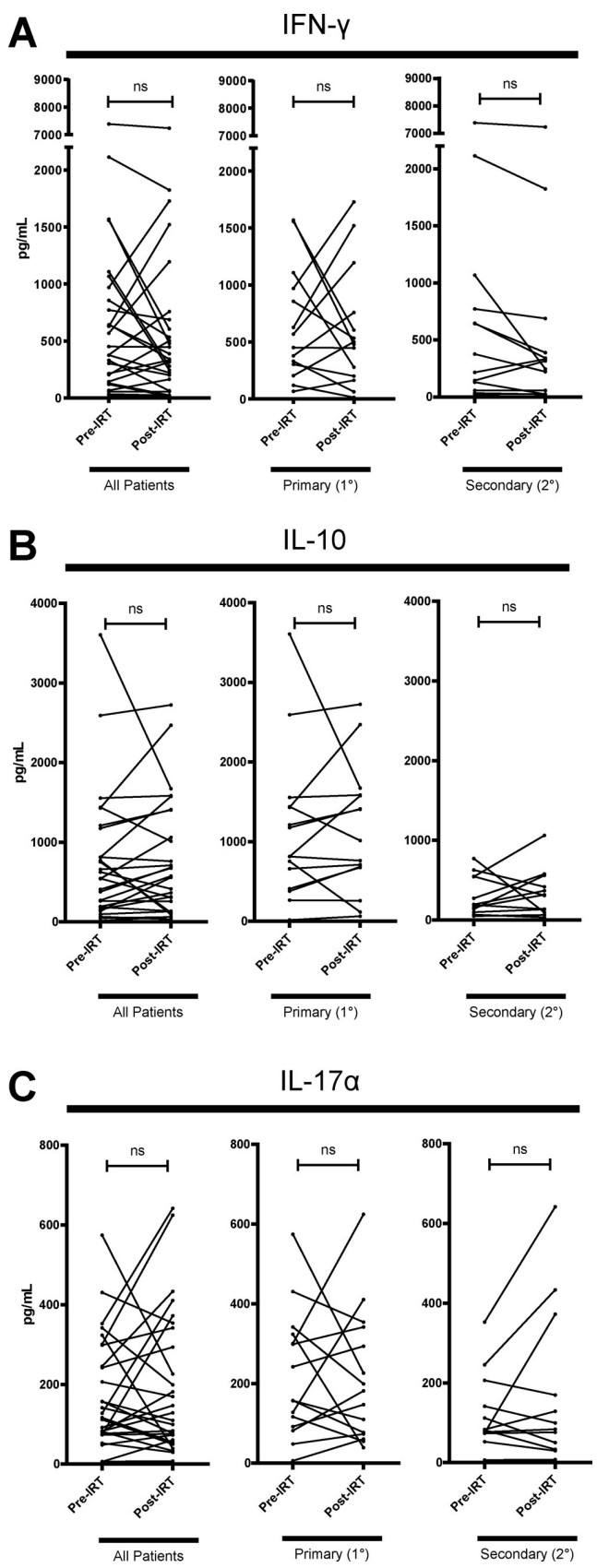

**Fig 2. IFN-γ, IL-10, and IL-17α cytokine production of α-CD3/CD28 stimulated PBMC supernatant.** Patient PBMCs were stimulated with α-CD3/CD28 and supernatant was analyzed for cytokine detection. (A) IFN-γ, (B) IL-10, and (C) IL-17α in supernatant were detected by cytometric bead array (CBA). Significant detectable quantities of IFN-γ were achieved after 6hr stimulation while IL-10 and IL-17α is detected in significant quantities after 48hr stimulation. All patients are shown and sub-stratified into primary (1˚) (n = 17) and secondary (2˚) (n = 14) antibody deficiency. IFN-γ expression in whole PBMC supernatant is significantly decreased in SAD patients post-IRT but not in PAD. Other cytokines reveal no significant change in expression post-IRT in both PAD and SAD patients. ns denotes not significant ($p > 0.05$), * denotes $p < 0.05$. P-values were determined by Wilcoxon matched-pairs signed rank test.

## Asymmetrical inhibition of T-cell proliferation by IRT in PAD and SAD

To measure proliferative capacity of T-cells upon stimulation, proliferation of CD3, CD4, and CD8 T-cells after 4 days of α-CD3/CD28 stimulation was measured via CellTrace dye dilution (Fig 5A). Decreased T-cell proliferation in both PAD and SAD patients was observed post-IRT. However, while all patients exhibited a significant decrease in general CD3 T-cell proliferation ($p = 0.025$) (Fig 5B), deeper analysis into T-cell subset revealed the decreased proliferation in CD4 T-cell compartment in PAD ($p = 0.025$) (Fig 5C) while the decreased proliferation was found in the CD8 T-cell compartment in SAD patients ($p = 0.042$) (Fig 5D). These results suggest that behind suppressed CD3 T-cell proliferation, proliferation of CD4 and CD8 T-cells is differentially modulated in context of underlying conditions between PAD and SAD patients post-IRT.

## Discussion

Our study reports proportion, cytokine production and proliferation of T-cells in patients with PAD and SAD before and after the initiation of IRT to explore the effect of IRT on T-cell immunity in these patient populations. We found that there was no change in resting CD4, CD8, Treg, or memory T-cell proportion in the PBMCs post-IRT. Notably, T-cells exhibited a stronger response to stimuli with increased IFN-γ producing CD4 and CD8 T-cells post-IRT as compared to baseline in SAD. Nevertheless, T-cell stimulation with MHC-I dependent CEF peptides did not reveal increased frequency of IFN-γ producing CD8 T-cells. Furthermore, total T-cell proliferation was decreased upon T-cell stimulation, but the reduction of T-cell proliferation was primarily due to reduced CD4 T-cell proliferation in PAD while it was primarily due to reduced CD8 T-cell proliferation in SAD patients.

The lack of significant differences in resting Treg proportion pre-IRT and post-IRT in our study is consistent with the prior work of Paquin-Proulx et al. who instead used whole blood analysis [12]. They found that short term IRT can transiently increase Treg in CVID patients but long-term IRT does not restore the deficient and dysfunctional Treg population. Previous studies with whole blood T-cell quantification described reduced CD4 T-cells, antigen-specific CD4 T-cells, particularly naïve CD4 T-cells in CVID [12, 41–44]. Furthermore, a strong correlation between the number of naïve CD4 T-cells and clinical severity in CVID has been demonstrated and that IRT improved deficient CD4 T-cell counts [12, 41–43]. Inconsistent with this literature, we report no change in CD4 T-cell and memory T-cell proportion post-IRT in PAD patients. It is conceivable that the PAD patient cohort in our study having milder phenotypes results in a less defective T-cell compartment. Less than a third of our PAD patient cohort had a CVID autoimmunity phenotype (Table 1) which was previously reported to be associated with a higher degree of T-cell defects, while the remaining patients were of a milder CVID or IgG subclass deficiency with milder alterations to T-cell compartments [35]. Additionally, our results extend this lack of long-term T-cell subset change to SAD patients.

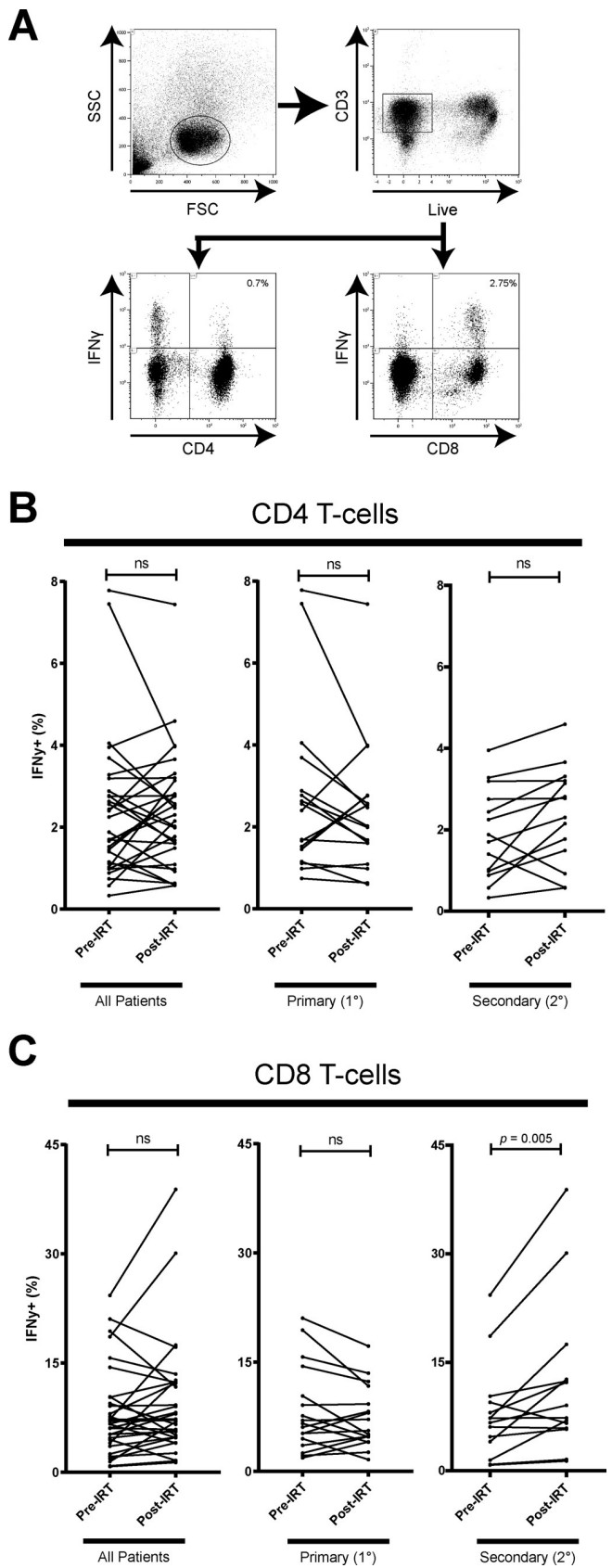

**Fig 3. Frequencies of IFN-γ+ CD4 and CD8 T-cells of α-CD3/CD28 stimulated PBMCs pre- and post-IRT.** (A) Patient PBMCs were stimulated with α-CD3/CD28 for 6hr in the presence of BFA and stained for CD3, CD4, CD8, and IFN-γ. Gating strategy for IFN-γ$^+$ CD4$^+$ and CD8$^+$ T-cells in stimulated PBMC is shown. (B) CD4$^+$ and CD8$^+$ T-cell IFN-γ expression after α-CD3/CD28 stimulation post-IRT is significantly higher in only 2˚ antibody deficiency patients, while 1˚ antibody deficiency patients trend downwards non-significantly. IFN-γ expression is detected in significant quantities after 6hr stimulation, and is more highly expressed in CD8 T-cells than CD4 T-cells. Patients are stratified into 1˚ (n = 17) and 2˚ (n = 14) antibody deficiency. ns denotes not significant ($p > 0.05$), $^*$ denotes $p < 0.05$, and $^{**}$ denotes $p < 0.01$. P-values were determined by Wilcoxon matched-pairs signed rank test.

Previous *in vitro* studies of IRT in healthy PBMC report downregulation of IL-2, IL-10, and IFN-γ cytokine production after α-CD3 stimulation, in addition to downregulation of inflammatory IL-17α production by T$_h$17 cells [25, 38]. Some of these studies are limited as the cytokine analysis did not differentiate between immune cells of origin. While our whole PBMC cytokine analysis did not reveal any significant trends for IFN-γ, IL-10, or IL-17α production post-IRT, intracellular staining of CD4 and CD8 T-cells reveal a stronger IFN-γ cytokine response upon α-CD3/CD28 stimulation post-IRT initiation than pre-IRT in SAD patients. Therefore, other IFN-γ-producing mononuclear cells such as NK cells may have masked changes in IFN-γ secretion from bulk PBMCs upon α-CD3/CD28 stimulation. This study was not set out to examine the role of other mononuclear cells, hence we could not explore the degree that these cells contribute to IFN-γ production. This increase in the IFN-γ+ CD4 and CD8 T-cell proportion suggests that IRT may enhance relative T-cell function acting in the pro-inflammatory cell-mediated T$_h$1 response, making SAD patients better able to respond to infection through IRT immunomodulation. On the other hand, overall IFN-γ production from PBMC supernatant of PAD patients on average did not change, there were heterogeneous results within the group and trend toward reduced IFN-γ+ CD4 and CD8 T-cell proportion in PAD after α-CD3/CD28 stimulation. This is consistent with the anti-inflammatory effects of IRT, although it is not significant. While the anti-inflammatory effects of IRT have been fairly well described, there still exists conflicting data against this effect, as one particular study studying immediate IVIG infusion in CVID patients reported higher expression of inflammatory cytokines IL-2 and TNF-α but not IFN-γ in CD4 and CD8 T-cells, suggesting that there may be a more nuanced mechanism at play [27]. The same trends of IFN-γ expression in α-CD3/CD28 stimulated CD4 and CD8 T-cells in PAD and SAD were not found when the memory CD8 T-cell population was stimulated with common virus-specific CEF peptides, even though drastic changes in cytokine production were observed in several patients. Since the CEF peptides represent common and/or opportunistic virus infections to which antibody deficient patients are highly susceptible, it is conceivable that the perturbed cytokine response might be influenced by recent experience to one of those infections. Although most patients in our cohort had past EBV infection, CMV seroprevalence was low compared to the approximately 44% reported in the general Canadian population [45]. It is not clear whether this was due to an insensitive serology assay to detect a low level of anti-CMV IgG, which was possible in these cases of hypogammaglobulinemia, or due to a truly low CMV infection rate in our cohort. A future study to examine the CD8 T-cell cytotoxic function in SAD patients post-IRT is warranted to further elucidate the possibility of IRT positively modulating CD8 T-cell function. Nonetheless, our results indicate that IRT recovers cytokine response of bulk T-cells without influencing memory T-cell compartment in SAD patients.

Our reported anti-proliferative effects of IRT on T-cells after α-CD3/CD28 stimulation highlight another asymmetry in T-cell subset affected between PAD and SAD patients receiving IRT. Although general CD3 T-cell proliferation was decreased, asymmetric anti-proliferative effects of IRT were observed in the CD4 T-cell subset in PAD and the CD8 T-cell subset in SAD. These results are somewhat consistent mechanistically with prior literature

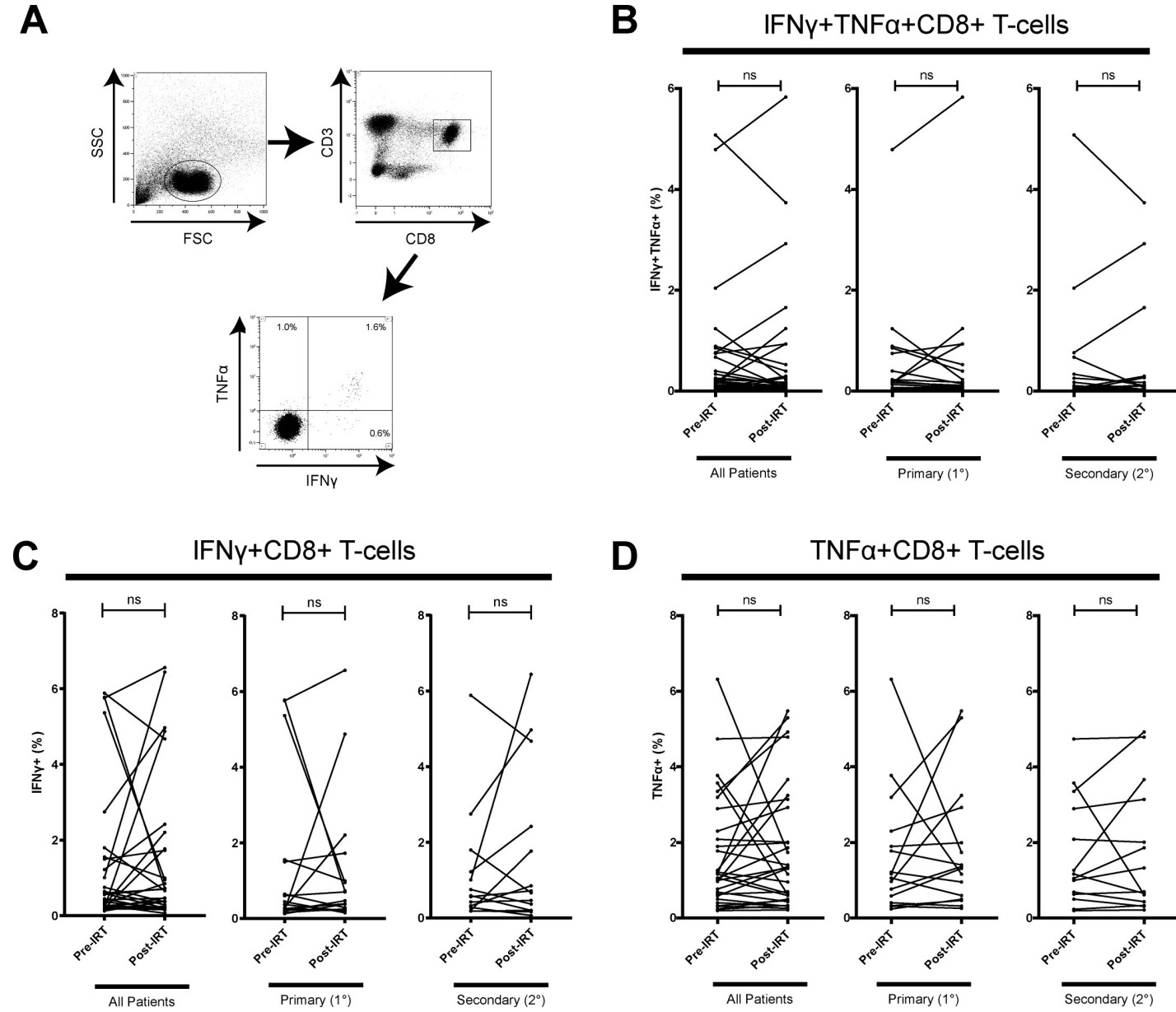

**Fig 4. IFN-γ and TNF-α expression after CEF peptide stimulation in CD8 T-cells from PAD and SAD patients pre- and post-IRT.** (A) Patient PBMCs were stimulated with CEF peptides for 6hr in the presence of BFA and stained for CD3, CD8, IFN-γ, and TNF-α. Gating strategy for IFN-γ+TNF-α+ CD8+ T-cells in CEF combo peptide stimulated PBMC is shown. (B) IFN-γ and TNF-α double-expression of memory CD8+ T-cells after CEF combo peptide stimulation does not significantly change post-IRT in either 1° or 2° patients. (B) IFN-γ and (C) TNF-α single positive expression of memory CD8+ T-cells after CEF combo peptide stimulation does not significantly change post-IRT in either 1° or 2° patients. Single positive expression of these cytokines in CEF combo peptide stimulation is weaker than from α-CD3/CD28 stimulation. Additionally, the magnitude of IFN-γ+TNF-α+ CD8 T-cells is very low. Patients are stratified into 1° (n = 17) and 2° (n = 14) antibody deficiency. ns denotes not significant (*p* > 0.05). *P*-values were determined by Wilcoxon matched-pairs signed rank test.

investigating the anti-proliferative effects of IVIg *in vitro*. Various groups using anti-CD3, PMA/ionomycin, or *Candida* antigen stimulations of healthy T-cells with added Ig in culture also report anti-proliferative effects in lymphocytes, but do not delve into T-cell subsets [46–48]. Independent of other immune cells, a mechanism of endogenous IgG-mediated regulation of T-cell activation and proliferation has been elucidated and may explain the observed decrease in T-cell proliferation *in vitro* and in our hypogammaglobulinemia patients as

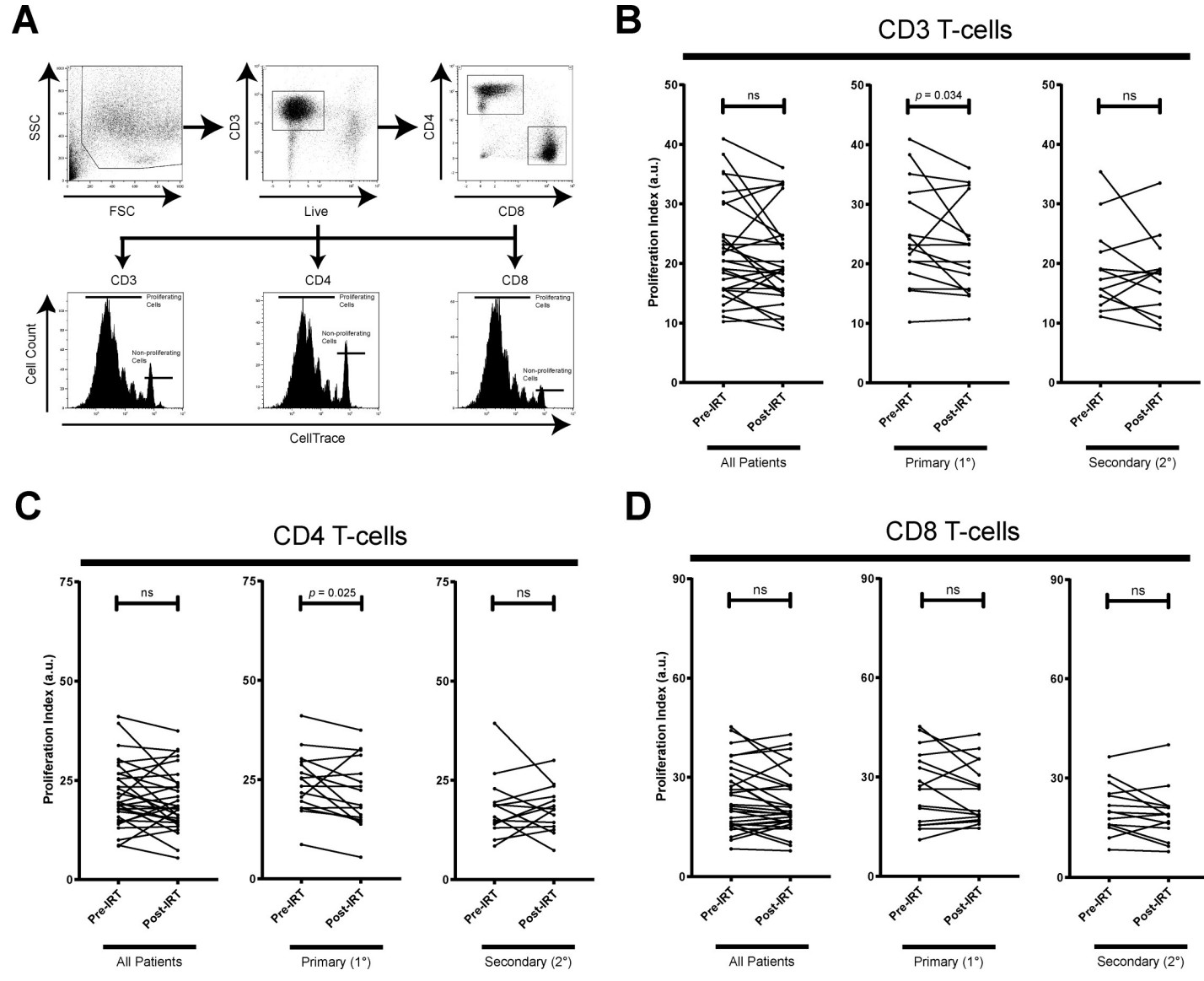

**Fig 5. T-cell proliferation after α-CD3/CD28 stimulation.** (A) Patient PBMCs dyed with CellTrace were stimulated with α-CD3/CD28 for 4 days stained for CD3, CD4, and CD8. Gating strategy for proliferating CellTrace dye-diluted CD3[+], CD4[+], and CD8[+] T-cells in PBMC is shown. CD3[+] T-cell proliferation is decreased post-IRT in all patients (C), but primarily due as decreased CD4[+] T-cell proliferation in 1˚ antibody deficiency patients (B) and primarily due as decreased CD8[+] T-cell proliferation in 2˚ antibody deficiency patients (D). Patients are stratified into 1˚ (n = 17) and 2˚ (n = 14) antibody deficiency. * denotes $p < 0.05$, and ** denotes $p < 0.01$. P-values were determined by Wilcoxon matched-pairs signed rank test.

presented here [47]. Additionally, anti-proliferative mechanisms involving IRT-mediated suppressive Treg expansion and impaired T-cell priming due to reduced antigen presentation by dendritic cells have also been underlined and could contribute to our observed results [49–52]. The differences in T-cell cytokine production in tandem with differences in T-cell proliferation between PAD and SAD patients post-IRT suggests that key differences in the underlying causes of immunodeficiency-linked hypogammaglobulinemia may bias the immunomodulatory effects of IRT towards certain T-cell subsets. It is also noteworthy that the presence of Ig from IRT in patient sera has been effectively eliminated in these T-cell functional assays during the PBMC processing and culture methods, suggesting that IRT can have longer lasting

immunomodulatory effects independent of the immediate presence of Ig. As this study is the first of its kind to report the anti-proliferative effect of IRT *ex vivo* in PAD and SAD patients without the persistent presence of Ig in culture, these results further support the proposed anti-inflammatory and "cooling down" effects of IRT on T-cell state and function.

A key limitation of this study is the lack of healthy controls to which pre-IRT patients can be compared in order to most strongly assess the perturbations in T-cell compartments pre-IRT. Without healthy controls, we decided to refer to literature values for normal ranges of T-cell subset proportions and CD4/CD8 ratio for a speculative look. Another limitation is the heterogeneity in the patient population regarding underlying conditions, medications, and comorbidities which may play a role in the observed T-cell function and trends (Table 1). In addition, it is possible that SAD patients exhibit changes in their T-cell compartment over time regardless of IRT. Nonetheless, this study's main focus and strength is the effect of IRT itself, thus we are able to analyze and observe changes in T-cell compartment and function post-treatment in the patient population.

In conclusion, our study reveals differential immunomodulatory effects of IRT on T-cells between patients with PAD and SAD. Our data also support the anti-inflammatory and anti-proliferative immunomodulatory effects of long-term IRT in PAD patients but paradoxically appear to enhance pro-inflammatory IFN-γ production in CD4 and CD8 T-cells upon α-CD3/CD28 stimulation for SAD patients alone. This may describe an enhanced T-cell functional recovery to produce IFN-γ post-IRT that renders SAD patients less vulnerable to common pathogens, leading to a reduced state of activation and proliferation. Trends observed in the SAD patients are especially noteworthy due to the high degree of heterogeneity of the patient population and the lack of work previously done in this patient population. While IRT remains a safe and effective treatment for hypogammaglobulinemia, these differences in their underlying cause and how IRT may mechanistically modulate certain subsets of T-cells more than others needs to be further explored.

## Supporting information

**S1 Fig. Comparison of pre- and post-IRT Tregs population proportions of unstimulated PBMCs from PAD patients with or without autoimmune disease.** Thawed cryopreserved PAD patient PBMCs were stained for CD3, CD4, CD25, CD127, and FOXP3 for Tregs (CD3$^+$CD4$^+$CD25$^{hi}$CD127$^{lo}$FOXP3$^+$). PAD patients (left panel) were subdivided based on either presence (center panel) or absence (right panel) of autoimmune disease. For each group, the proportion of Tregs among CD4$^+$ T-cells were compared between pre- and post-IRT. Ns denotes not significant ($p > 0.05$). *P*-values were determined by Wilcoxon matched-pairs signed rank test.
(TIFF)

## Acknowledgments

We thank Lynda Theoret, R.N., for her help with patient care, IRT administration, and blood sample drawing. We also thank the patients who graciously donated their time and energy to this study. We thank the National Institutes of Health/National Cancer Institute at Frederick Biological Resources Branch Preclinical Repository for providing the CEF peptides.

## Author Contributions

**Conceptualization:** Tri Dinh, Jun Oh, Donald William Cameron, Seung-Hwan Lee, Jutha-porn Cowan.

**Formal analysis:** Tri Dinh.

**Funding acquisition:** Seung-Hwan Lee, Juthaporn Cowan.

**Investigation:** Tri Dinh, Jun Oh.

**Methodology:** Tri Dinh, Jun Oh, Donald William Cameron, Seung-Hwan Lee, Juthaporn Cowan.

**Project administration:** Donald William Cameron, Seung-Hwan Lee, Juthaporn Cowan.

**Resources:** Tri Dinh, Jun Oh, Seung-Hwan Lee, Juthaporn Cowan.

**Supervision:** Donald William Cameron, Seung-Hwan Lee, Juthaporn Cowan.

**Validation:** Tri Dinh.

**Visualization:** Tri Dinh.

**Writing – original draft:** Tri Dinh, Jun Oh, Donald William Cameron, Seung-Hwan Lee, Juthaporn Cowan.

**Writing – review & editing:** Tri Dinh, Jun Oh, Seung-Hwan Lee, Juthaporn Cowan.

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
