## [Decision Letter · Decision Letter 0]

26 Jul 2019

PONE-D-19-14805

Differential immunomodulation of T-cells by immunoglobulin replacement therapy in primary and secondary antibody deficiency

PLOS ONE

Dear Dr Cowan,

Thank you for submitting your manuscript to PLOS ONE. After careful consideration, we feel that it has merit but does not fully meet PLOS ONE’s publication criteria as it currently stands. Therefore, we invite you to submit a revised version of the manuscript that addresses the points raised during the review process.

The low number of patients, different route of immunoglobulin administration, and the heterogeneity of PID and SID  are the major concerns.

We would appreciate receiving your revised manuscript by Sep 09 2019 11:59PM. To enhance the reproducibility of your results, we recommend that if applicable you deposit your laboratory protocols in protocols.io, where a protocol can be assigned its own identifier (DOI) such that it can be cited independently in the future. For instructions see: http://journals.plos.org/plosone/s/submission-guidelines#loc-laboratory-protocols

We look forward to receiving your revised manuscript.

Kind regards,

Jagadeesh Bayry, DVM, PhD, HDR

Academic Editor

PLOS ONE

**Journal Requirements:**

"Juthaporn Cowan received research funds and honoraria from CSL Behring, Grifols, Shire and Octapharma, outside the submitted work. "

**Comments to the Author**

1. Is the manuscript technically sound, and do the data support the conclusions?

Reviewer #1: Partly

Reviewer #2: No

2. Has the statistical analysis been performed appropriately and rigorously? 

Reviewer #1: Yes

Reviewer #2: Yes

3. Have the authors made all data underlying the findings in their manuscript fully available?

Reviewer #1: Yes

Reviewer #2: Yes

4. Is the manuscript presented in an intelligible fashion and written in standard English?

Reviewer #1: Yes

Reviewer #2: Yes

5. Review Comments to the Author

Reviewer #1: The manuscript attempts to study and understand the immune modulation mechanisms for immunoglobulin replacement therapy in PAD and SAD. The manuscript is well-written with ample discussion. It addresses an important topic on benefits of IRT in different situations of immune deficiency. Although an interesting observation, there are several conceptual and technical concerns that precludes publication in the present state. The authors need to provide more information on these for better clarity.

Specific Comments

1. Are the PAD patients enrolled in the study recently diagnosed or were previously treated with IRT before? As per the table 1, no previous biologics treatment was received in PAD. Please confirm.

2. Information on the time between first diagnosis to initiation of IRT, and any previous immune modulatory or antibody replacement therapies for PAD patients, need to be provided to help understand the impact of any previous treatment versus current IRT.

3. Table 2. Include a column beside T-cell subset to depict the unit of expression as % of PBMCs, % of CD4+ T cells or % of CD3+ T cells. Remove top row, proportion of PBMCs.

4. Figure.1, there is a trend towards increase in Treg post-IRT in PAD. Is the increase mainly noted in PAD patients with autoimmune disease? Did authors compare Treg functionality pre and post IRT in T-cell proliferation assay?

5. Since there are multiple data sets for each donor pre and post-IRT and for each parameter there are donors responding differently post-IRT. Therefore, a correlation analysis could be done to see which parameters correlate positive versus negatively. Also, normalized results for post-IRT for different parameters could be plotted as a heatmap to give an overview of changes due to IRT in PAD and SAD patients.

6. Figure 4 and 5, in PAD patients the trend of decreased IFN-g+ T cells and proliferation is noted. Did PAD with autoimmune response showed higher change than others?

7. Ref 38 citation in discussion is not correct. This study used purified CD4+ T cells to show high-dose IVIG inhibition of IL-17 and IFN-g production. Also, paper showed that low concentration of IVIG does not interfere with IL-17 secretion. Please state this with reference.

8. Authors note in the discussion that IRT might influence Treg suppressive function. Please include these relevant papers on mechanisms of Treg modulation by IRT (IVIG). Pubmed ID: 28916232, 28481908, 28284485, 25391612.

9. Authors need to provide more results or explanation on the reason for differential modulation of T-cells in PAD and SAD following IRT.

Reviewer #2: In the manuscript "Differential immunomodulation of T-cells by immunoglobulin replacement therapy in primary and secondary antibody deficiency" the Authors assessed modulation of T-cell function by immunoglobulin replacement therapy in a small group of patients affected by primary and secondary antibody deficiencies. The low number of patients, the different route of immunoglobulin administration, and the heterogeneity of PID and SID diagnoses did not allow to make any understandable conclusion. Moreover, additional treatments other than immunoglobulin replacement might have strongly influenced the results.

6. PLOS authors have the option to publish the peer review history of their article (what does this mean?). If published, this will include your full peer review and any attached files.

Reviewer #1: No

Reviewer #2: No

---

## [Author Response · Author response to Decision Letter 0]

10 Sep 2019

Reviewer #1: The manuscript attempts to study and understand the immune modulation mechanisms for immunoglobulin replacement therapy in PAD and SAD. The manuscript is well-written with ample discussion. It addresses an important topic on benefits of IRT in different situations of immune deficiency. Although an interesting observation, there are several conceptual and technical concerns that precludes publication in the present state. The authors need to provide more information on these for better clarity.

Thank you for your comments. Please find point-by-point responses to your concerns below. 

Specific Comments

1. Are the PAD patients enrolled in the study recently diagnosed or were previously treated with IRT before? As per the table 1, no previous biologics treatment was received in PAD. Please confirm.

All except one PAD patients enrolled in the study were newly diagnosed and were not previously treated with IRT. The one patient who was diagnosed more than 10 years ago was on IRT for a few months at the time of diagnosis. Therefore, at the time of study enrolment, the patient had not been on IRT for over 10 years. 

Yes, it is correct that no previous biologics treatment was given to PAD patients except one who had rituximab treatment six years prior to this study enrolment. 

This information is clarified in the study inclusion criteria and table 1. 

2. Information on the time between first diagnosis to initiation of IRT, and any previous immune modulatory or antibody replacement therapies for PAD patients, need to be provided to help understand the impact of any previous treatment versus current IRT.

Thank you for pointing this out. The average time from the first diagnosis to initiation of IRT was 1.6 months (SD 1.5). This number excluded the one patient who had initial diagnosis more than 10 years ago. This information was added in the result section page 9. 

3. Table 2. Include a column beside T-cell subset to depict the unit of expression as % of PBMCs, % of CD4+ T cells or % of CD3+ T cells. Remove top row, proportion of PBMCs.

Thank you. Table 2 was amended as suggested. 

4. Figure.1, there is a trend towards increase in Treg post-IRT in PAD. Is the increase mainly noted in PAD patients with autoimmune disease? Did authors compare Treg functionality pre and post IRT in T-cell proliferation assay?

This is an excellent point. We have examined the trend in the group of PAD patients with autoimmune disease. The trend towards increase in Treg post-IRT in PAD was not observed in PAD patients with autoimmune disease. However, there were only 5 PAD patients with autoimmune disease. We included the data as a supplementary figure. 

Unfortunately, we did not have adequate samples to compare Treg function pre and post IRT in regard to cell proliferation. Such testing would require a large number of T cells. It would be interesting to compare differences in Treg number and function post IRT in PAD patients with and without history of autoimmunity. Such study would be successful in a reasonable time in a multicenter study. 

5. Since there are multiple data sets for each donor pre and post-IRT and for each parameter there are donors responding differently post-IRT. Therefore, a correlation analysis could be done to see which parameters correlate positive versus negatively. Also, normalized results for post-IRT for different parameters could be plotted as a heatmap to give an overview of changes due to IRT in PAD and SAD patients.

We agree that the identification of associated factors or predictors for changes in T-cell post IRT would have been informative. We discussed this with a biostatistician. The challenge was that the sample size was too small to draw any meaningful conclusion from such analysis. We agree with you that sample size is our major limitation. However, it is the largest cohort to date with comparison to SAD population. 

6. Figure 4 and 5, in PAD patients the trend of decreased IFN-g+ T cells and proliferation is noted. Did PAD with autoimmune response showed higher change than others?

Similar to point number 4, we did not see a significant reduction of IFN-g+T cells and proliferation with the subgroup analysis. 

7. Ref 38 citation in discussion is not correct. This study used purified CD4+ T cells to show high-dose IVIG inhibition of IL-17 and IFN-g production. Also, paper showed that low concentration of IVIG does not interfere with IL-17 secretion. Please state this with reference.

Thank you for your thorough review. We corrected the sentence. It now reads “Downregulation of inflammatory IL-17α production by Th17 cells was also seen after high dose but not low dose Ig [38].”

8. Authors note in the discussion that IRT might influence Treg suppressive function. Please include these relevant papers on mechanisms of Treg modulation by IRT (IVIG). Pubmed ID: 28916232, 28481908, 28284485, 25391612.

Thank you kindly for these references. We added them into our discussion on page 22. 

9. Authors need to provide more results or explanation on the reason for differential modulation of T-cells in PAD and SAD following IRT.

Thank you. We added a couple of sentences to our discussion regarding our thoughts on differential modulation of T-cells on page 21. It reads “Alternatively, many SAD patients in our cohort were on immunosuppressive and immunomodulating agents that in itself could have temporally modulated T-cell function regardless of IRT. A control group of SAD patients who did not receive IRT would have provided additional insight on this speculation”.

Reviewer #2: In the manuscript "Differential immunomodulation of T-cells by immunoglobulin replacement therapy in primary and secondary antibody deficiency" the Authors assessed modulation of T-cell function by immunoglobulin replacement therapy in a small group of patients affected by primary and secondary antibody deficiencies. The low number of patients, the different route of immunoglobulin administration, and the heterogeneity of PID and SID diagnoses did not allow to make any understandable conclusion. Moreover, additional treatments other than immunoglobulin replacement might have strongly influenced the results.

Thank you for your review and comments. We agree that the major limitation of this study is the low number of patients. However, to our knowledge, our PAD cohort is the largest to date with respect to assessing the modulatory effect of IRT in humans. In addition, comparative study on modulatory effect of IRT between PAD and SAD was not published previously. We feel that although a definitive conclusion cannot be made from our findings. It is served as a baseline for further studies. 

Route of Ig administration is not a significant limitation of this study in our view as only one patient was on IVIG while the rest were on SCIG. In addition, the post-IRT blood sample was drawn several months after IRT initiation to ensure the steady state of IRT with plateau IgG trough level. The equivalent clinical efficacy of IVIG and SCIG has been shown previously in patients with antibody deficiencies. 

The reviewer is absolutely right that other treatments than IRT may have influenced the results, and we think that this could be one of the reasons why there is differential effect of IRT between PAD and SAD patients. PAD patients in our cohort were quite uniform in that they were not on immunosuppressive or immunomodulating agents, unlike many SAD patients in our cohort.

---

## [Decision Letter · Decision Letter 1]

1 Oct 2019

Differential immunomodulation of T-cells by immunoglobulin replacement therapy in primary and secondary antibody deficiency

PONE-D-19-14805R1

Dear Dr. Cowan,

We are pleased to inform you that your manuscript has been judged scientifically suitable for publication and will be formally accepted for publication once it complies with all outstanding technical requirements.

With kind regards,

Jagadeesh Bayry, DVM, PhD, HDR

Academic Editor

PLOS ONE

Additional Editor Comments (optional):

Reviewers' comments:

Reviewer's Responses to Questions

**Comments to the Author**

1. If the authors have adequately addressed your comments raised in a previous round of review and you feel that this manuscript is now acceptable for publication, you may indicate that here to bypass the “Comments to the Author” section, enter your conflict of interest statement in the “Confidential to Editor” section, and submit your "Accept" recommendation.

Reviewer #1: All comments have been addressed

2. Is the manuscript technically sound, and do the data support the conclusions?

Reviewer #1: Yes

3. Has the statistical analysis been performed appropriately and rigorously? 

Reviewer #1: Yes

4. Have the authors made all data underlying the findings in their manuscript fully available?

Reviewer #1: Yes

5. Is the manuscript presented in an intelligible fashion and written in standard English?

Reviewer #1: Yes

6. Review Comments to the Author

Reviewer #1: The authors have addressed all the concerns satisfactorily by providing additional information and including them in tables and text. The study helps to understand the immune modulation

mechanisms for immunoglobulin replacement therapy in PAD and SAD.

7. PLOS authors have the option to publish the peer review history of their article (what does this mean?). If published, this will include your full peer review and any attached files.

Reviewer #1: No

---

## [Editor Report · Acceptance letter]

4 Oct 2019

PONE-D-19-14805R1 

Differential immunomodulation of T-cells by immunoglobulin replacement therapy in primary and secondary antibody deficiency 

Dear Dr. Cowan:

I am pleased to inform you that your manuscript has been deemed suitable for publication in PLOS ONE. Congratulations! Your manuscript is now with our production department. 

With kind regards,

on behalf of

Dr. Jagadeesh Bayry 

Academic Editor

PLOS ONE